# Barriers and Facilitators of Safe Communication in Obstetrics: Results from Qualitative Interviews with Physicians, Midwives and Nurses

**DOI:** 10.3390/ijerph18030915

**Published:** 2021-01-21

**Authors:** Martina Schmiedhofer, Christina Derksen, Franziska Maria Keller, Johanna Elisa Dietl, Freya Häussler, Reinhard Strametz, Ilona Koester-Steinebach, Sonia Lippke

**Affiliations:** 1Department of Psychology & Methods, Jacobs University Bremen GmbH, Campus Ring 1, 28759 Bremen, Germany; C.Derksen@jacobs-university.de (C.D.); f.keller@jacobs-university.de (F.M.K.); dietl@aps-ev.de (J.E.D.); s.lippke@jacobs-university.de (S.L.); 2German Coalition for Patient Safety (Aktionsbuendnis Patientensicherheit e.V.), Alte Jakob Str. 81, 10179 Berlin, Germany; haeussler@aps-ev.de (F.H.); reinhard.strametz@hs-rm.de (R.S.); koester-steinebach@aps-ev.de (I.K.-S.); 3Wiesbaden Business School, Rhein Main University of Applied Science, 65183 Wiesbaden, Germany

**Keywords:** patient safety, interprofessional communication, obstetrics, preventable adverse events, interprofessional cooperation, safety II, qualitative health research

## Abstract

Patient safety is an important objective in health care. Preventable adverse events (pAEs) as the counterpart to patient safety are harmful incidents that fell behind health care standards and have led to temporary or permanent harm or death. As safe communication and mutual understanding are of crucial importance for providing a high quality of care under everyday conditions, we aimed to identify barriers and facilitators that impact safe communication in obstetrics from the subjective perspective of health care workers. A qualitative study with 20 semi-structured interviews at two university hospitals in Germany was conducted to explore everyday perceptions from a subjective perspective (subjective theories). Physicians, midwives, and nurses in a wide span of professional experience and positions were enrolled. We identified a structural area of conflict at the professional interface between midwives and physicians. Mandatory interprofessional meetings, acceptance of subjective mistakes, mutual understanding, and debriefings of conflict situations are reported to improve collaboration. Additionally, emergency trainings, trainings in precise communication, and handovers are proposed to reduce risks for pAEs. Furthermore, the participants reported time-constraints and understaffing as a huge burden that hinders safe communication. Concluding, safety culture and organizational management are closely entwined and strategies should address various levels of which communication trainings are promising.

## 1. Introduction

Patient safety can be defined as the “absence of preventable harm to a patient during the process of health care and reduction of risk of unnecessary harm” [1,2] and is an important objective in health care system [3]. Prevention of adverse events (AEs) is essential for patient safety. AEs are defined as temporary or permanent disability, death, or prolonged hospital stay. While some AEs are not preventable, patient safety has to focus on preventable adverse events (pAEs) defined as a harmful result of the care that fell below the standard expected in healthcare settings [4]. The understanding of the burden from errors in medical care has increased globally during the last few decades. [5]. Since the Institute of Medicine’s report “To err is human” was published in the year 2000, patient safety was given its own agenda with the aim to enhance good quality in health care. To take this aim forward, creating a safety culture on all levels of health care is necessary. Data suggest that inpatient treatment in countries like Germany is associated with an incidence of pAEs between 2 and 4 percent and avoidable mortality is assumed to be at 0.1 percent of all inpatients [6,7]. Framed positively, one may conclude, that in 96 to 98 percent of all cases, the treatment is delivered perfectly. Still, the aim needs to be to prevent all possible pAEs.

By all means, human factors play an important role in developing and implementing improved patient safety structures in everyday life, and can be captured by subjective theories of professionals. To counter the sources for errors, strategies have been implemented such as mandatory reporting about patient safety incidents (e.g., NHS never event framework) [8,9]. However, to implement strategies to improve safety, safety management needs not only to ask why things go wrong but also to acknowledge that most of the care works well, as clinicians adjust their work to real-life conditions. In addition, to avoid a pAE, it could be looked at how and why care is delivered well in most cases [10]. This “work-as-done” approach focuses on tasks under given working conditions instead of “work-as-imagined”, which tends to ignore obstacles staff has to cope with in everyday life [11]. The approach covers both, how adverse events were triggered and how safety could be improved based on quality-based knowledge. Nonetheless, health care systems are complex organizations, whose outcomes are determined by financing structures, occupational training, workload, and professional regulation [12]. While the patient perspective is a crucial indicator of quality problems and improvements in health care, health care providers are working more constantly in the setting and are held responsible if pAEs occur. Since they are the experts for their “work-as-done”, improvement strategies in the clinical setting, focusing on the health care providers are needed. This study was conducted as part of the “TeamBaby—safe, digitally supported communication in obstetrics” project which aims to avoid pAEs by implementing safe communication in different ways. Therefore, we started with subjective theories from the professional perspective on proximal determinants of patient safety and adverse events to base future improvement measures on the results. Our approach allows us to study patient safety from a so-called “safety I” perspective. This perspective aims to identify, understand, and overcome the sources of failure. In contrast, the “safety II” approach aims to reach a more positive perspective with understanding and improving how processes lead to good care in everyday clinical work [13]. One important determinant of patient safety is team collaboration and communication. As communication encompasses the essential ability not only to pass clear information, but also to reconsider and discuss challenging treatment decisions on team level, we conducted this study in a field associated with high risk for patients to understand communication and resilience in health care systems.

Although most of the women giving birth are healthy, obstetrics can be regarded to be a high-risk area [14]. The majority of women do not require classic treatment of chronic or acute disease, but a large number of women may need routine medical support during labor such as pain relief, anesthesia, treatment, or prevention of infections. However, the delivery process is hardly predictable, and situations may change instantly requiring medical emergency treatment. Moreover, the medical staff is always responsible for both, the mother and the newborn. In obstetric wards, midwives, residents, obstetricians, and nurses need to team up with anesthesiologists and pediatricians, often trained in contrasting styles regarding decision-making processes, consideration of patients’ preferences, speaking-up and communication skills. However, frequent personnel changes in large departments pose a significant challenge. In the context of obstetrics, pAEs include any physical or mental harm to the pregnant woman or newborn caused by incorrect or delayed medical decisions or actions. In a recent meta-analysis, the mean incidence of AE in gynecological hospital admissions was 10.5 percent of which about half could have been prevented and 1.2 percent led to death [15]. Hence, a better understanding of barriers (safety I) and facilitators (safety II) that impact safe communication in obstetrics is needed from the subjective perspective of health care workers.

Most non-physician health care professions, such as nurses or physiotherapists are dependent on doctor’s advice and directions in Germany (“Arztvorbehalt”). However, midwives are mandated to work autonomously most of the time focusing on positive outcomes, thus achieving safety II [16]. In case of complications during labor, they are required to consult a doctor leading to a shared decision process. This interface between these professional groups needs a high commitment of time-critical and precise transmission of information. Views on required decisions, e.g., on the timing or necessity of a cesarean section, may differ. Furthermore, the preferences of the women in labor have to be considered as much as possible, even in time-critical situations. In upcoming emergencies, the need for successful collaboration is particularly high but at the same time highly challenging, as the birth process may require rapid action and decision-making along with dynamic shifts in responsibilities. Therefore, effective and trustworthy (“safe”) communication matching the safety II approach is of major importance. Mutual respect and closed-loop communication contribute significantly to the reduction of pAEs [17]. Furthermore, enhanced provider communication leads to increased job satisfaction, which in turn has positive effects on patient care and patient satisfaction [18].

Taking this into account, the research objective was to investigate how safe communication can be achieved in the obstetric setting. To answer this question, the aim of this study was to identify barriers and facilitators that impact safe communication in obstetrics from the subjective perspective of health care workers by means of qualitative interviews with a heterogeneous group of 20 physicians, midwives and nurses. More specifically, the research question focuses on the everyday perceptions of the participating professions (subjective theories) regarding the objective to communicate safely and with that gathering learning proactively about safety issues beyond specific ones [19]. Specifically, the research objective is to identify barriers and facilitators of safe communication (as they are key to the subsequent implementation of successful improvement strategies and contribute significantly to resilient health care) by means of qualitative interviews to get more insights not observable with other methods [20].

## 2. Materials and Methods

This qualitative interview study is one module of the mixed-methods research project “TeamBaby”. Detailed information have been described during study registration (NCT03855735) and in the published study protocol [21]. The purpose of this intervention study is to contribute to the reduction of pAEs in obstetrics and thus to increase patient safety through improved communication skills. To reach this aim, clinical staff, expectant mothers, and their partners are trained in self-confident communication skills to overcome difficulties in everyday hospital life. In order to assess the subjective perception of interprofessional communication and to understand performance pattern, qualitative interviews were conducted with members of the included health care teams.

### 2.1. Study Design

A qualitative study with semi-structured, face-to-face interviews at two university hospitals in Germany was conducted. In Hospital I, there were about 2300 births per year and in Hospital II about 2800. We aimed to conduct a stratified purposive sampling and accordingly enrolled a purposive and heterogeneous sample of obstetric staff from both sites [22]. We focused on the core team of obstetrics, that is, all addressed physicians were obstetricians, either as residents, or consultants, or in a senior position. Participants were approached by research staff working in the clinics according to their profession and work experience. The aim of the project was introduced during several meetings and potential participants were addressed. To make participation as convenient as possible, interviewees were invited to choose the time and location of the interviews. The project was supported by the hospital management and was already known through several research activities. Therefore, there were no rejections to participate. To ensure data protection, the results obtained were analyzed and presented jointly.

Between December 2019 and April 2020, one of the authors (MS) conducted a total of 20 interviews at the two sites of which nine were carried out at site I and 11 at site II. A total of 19 interviews were performed face-to-face in person in separated rooms of the respective clinics. Due to Covid-19 restrictions, the last interview took place via an online platform. All subjects gave written informed consent before they participated in the study collection. Following each interview, field notes were taken to document impressions on atmosphere, nonverbal communication, and special features and used for better understanding and interpreting of the orally recorded interviews. The interviews took from 19 to 90 min with a median duration of 36 min. The interviews were audiotaped and then transcribed verbatim. The study was conducted in accordance with the Declaration of Helsinki, and the protocol was approved by the Ethics Committee of the University Clinics Ulm (114/19-FSt/Sta; 29 Mai 2019) and Frankfurt am Main (No. 19-292; 22 August 2019).

### 2.2. Interview

A semi-structured interview guide with open questions was developed to obtain in-depth, detailed insights into obstetrics staff’s perspectives (Table 1). The structure was based on the literature and the researcher’s knowledge of the subject [5,17,18]. Questions to generate suitable answers to the research question were developed through discussions of the interdisciplinary TeamBaby research group (MS, CD, JD, FH, FK, SL) including two Masters of Public Health (MPH) and four psychologists. The interview guide was modified after the first two interviews. During the interviews, the order of questions was adapted to the narrative flow and openness of the individual participants. Theoretical saturation was reached when new findings could no longer be obtained after 20 interviews at both hospital sites.

### 2.3. Participants’ Characteristics

The heterogeneous and stratified sample includes midwives, physicians, and nurses in a wide span from training to superior level positions. As shown in Table 2, their professional experience ranges from one to 31 years with a median of 9 and a mean of 12 years.

Stratified purposive sampling was accomplished although few men and no nurses with fewer than 5 years in the job (occupational age) could be recruited [22]. There were no refusals of potential interviewees to participate in the study.

### 2.4. Data Analysis

All interview transcripts and field notes were entered into the qualitative data software MAXQDA2020 and anonymized for analysis. A qualitative content analysis (QCA) approach was taken to work out the results, using a multi-stage process. QCA works equally inductively and deductively into themes emerging from text-analysis [23]. To answer the research question, one of the authors (MS) reviewed the transcripts and coded them line by line. Sentence chunks or single words were labeled with broad categorization. Then, the material was carefully re-read and completely recoded as new themes emerged. All steps were documented in a logbook. To make the coding process transparent for all team members, a spreadsheet with all codes and underlying quotations was built. In subsequent discussions, the multidisciplinary research group and participants of an evaluation methods seminar at Jacobs University Bremen (SL) refined the final code structure. Based on this structure, attitudes, perceptions, and explanation patterns were compared and contrasted between and within occupational groups. Then, JD and FH checked and revised the coding and assigned them, independently and separately, to the categories. Finally, the main categories were built to answer the research question. The consensus was sought (MS, JD, FH) until the agreement was achieved. The results are presented with significant quotes from the participants. The interviews conducted in German were first translated into English by MS and then checked for correctness by SL and CD.

## 3. Results

The data interpretation was based on the interviewees’ subjective views of strengths and weaknesses concerning interprofessional communication and cooperation. The results were triangulated by comparing the answers of all occupational groups [23]. In doing so, we present behavioral and explanatory patterns that impacted communication in everyday performance, where professional behavior was adapted to working conditions. Worth mentioning are the interdependences between professional experience and professional status. That is, on the one hand, young residents and experienced midwives worked together in a supportive manner and more equally than those with greater hierarchical gradients. On the other hand, it became clear that the transition to pathological birth courses needed a lot of attention and was at the time structurally conflictual between midwives and physicians who had to assume the main medical responsibility.

The results are presented in three sections, each contrasting and comparing the views of the professional groups. First, the organization and perception of the division of tasks in everyday work are described. In the second section, statements leading to safe communication are presented: the perception of speaking-up, the subjective handling of conflicts, and the perceived support in the work environment. In the third section, barriers and factors that hinder or promote safe communication are presented and assigned to three levels of analysis: The team (midi-level), the clinic (meso-level), and the health care system (macro-level). This structure corresponds to the statements of the participants, who identified several conditions affecting patient safety outside the team structure.

All results are documented with illustrative quotations. In Table 3 and Table 4, we present the perceived collaboration and communication from the different professional perspectives from everyday life experience (Table 3) as well as with the focus on conflicts (Table 4). The subjective theories about causes of pAEs and suggestions to avoid pAEs from our respondents are displayed in Figure 1. The corresponding quotes are in the appendix.

### 3.1. Task Sharing in Everyday Life

The share of tasks is outlined by all professions in conformity with the legal situation: Midwives are responsible for a physiological birth process and have to consult doctors in case of arising difficulties, who in turn can be legally prosecuted for treatment errors. Despite the fact that collaboration was largely described as cooperative and supportive, conflicting patterns of professional perceptions became visible, mainly in time-critical situations. The statements of physicians revealed how they put their clinical treatment path view above the assessment of midwives. If midwives did not agree with the physicians’ decision, they may have considered the approach as disrespectful. However, the degree of professional hierarchy influencing the decision-making process appeared linked to the length of professional experience: young residents reported how grateful they were to be trained by experienced midwives. Interestingly, midwives assumed that doctors were not deeply interested in participating in a physiological birth process while doctors described differing attitudes: Some were sorry to not have more time for the women, while others saw the supportive care as basic midwife tasks. The nursing staff on the ward contributed significantly to women’s well-being in the post-natal phase, even though they were not involved in the acute birthing process. In critical pregnancies, they could be important social support providers for a long period of time. In the interviews, it became clear that nurses external to the delivery room team sometimes felt not sufficiently appreciated by the delivery room staff, because they took over the care for the women in subacute situations. This became visible in reported conflicts about duty transmission, partly confirmed by midwives.

### 3.2. Managing Conflicts

Regarding conflicts and how to manage them, two areas were identified, namely speaking up in terms of expressing safety concerns from all levels of hierarchy and how to deal with uncertainty and conflicts with peer and leadership support. These two areas are described in the following.

#### 3.2.1. Speaking Up: Expressing Safety Concerns from All Levels of Hierarchy

As shown above and reported in Table 4 (upper part), there may have been differences in the assessment of risk situations between the occupational groups involved. Due to the relative unpredictability of birth events, professional experience played a major role in the safety of mother and newborn. The reported willingness to bring in a different opinion depended on factors such as personality, level of the hierarchy, personal relationships, and the assumption to be listened to by a superior. Again, different perspectives between midwives and physicians became visible. Additionally, stress and unclear presumptions about the intention of the acting persons impacted raising or waiving an objection. Interviewees highlighted that the willingness to raise concerns depended on the degree of confidence and trust within the team. That is, discussing mistakes and debriefing to a conflict without being afraid of social exclusion or accusations. Above all, residents and midwives in training emphasized the importance of being encouraged to admit uncertainties and ask questions without being afraid of negative reactions.

#### 3.2.2. Dealing with Uncertainty and Conflict: Peer and Leadership Support

Facing uncertainties and doubts is reported to be a major issue in everyday work, see Table 4 (lower part). The opportunity to ask questions even in stressful situations was described as an asset to overcome discomfort and an important encouragement for professional satisfaction, especially for staff undergoing training. In turn, a lack of support may lead to sustainable professional dissatisfaction and decreased patient safety. Facing conflicts that remained unresolved were reported as common experiences in daily work. Handlings were described in a range from coping with the issue at home silently, insisting on debriefing up to actively calling for management support. Addressing conflicts in the team and perceiving the back-up of superiors, especially in interprofessional conflicts, was highly appreciated and reported to be beneficial. Additionally, social support in coping with fatal incidents (as intrauterine fetal deaths or stillbirths) was described as helpful by young midwives. However, debriefing on distressing incidents seems to depend more on the personal and informal commitment of individuals involved than on official structures in the organization. Therefore, participants of all professions and at all levels of hierarchy sought for leadership’s responsibility to ensure trustful collaboration.

### 3.3. Subjective Theories about Causes of pAEs and Suggestions to Avoid pAEs

Even though the interview guide focused on personal experience, participants reported impacts outside of team to be accountable for pAEs. Examples for contributing factors corresponded with suggestions for improved communication and patient safety. In the following, they are outlined in order to the respective levels of responsibilities. Significant quotes are presented in the Appendix A and an overview is given in Figure 1.

In Figure 1, the subjective theories relating to the three areas team, hospital, and health care system clearly indicate that they could be distinguished but at the same time, they built on each other. In the following, the different layers are described in more detail to synthesize the interviews.

#### 3.3.1. Team Level

At the team level, lack of shared knowledge, insufficient transfer of information, or divergent perspectives on appropriate treatment were indicated to trigger pAEs. Accordingly, optimized structures and improved interpersonal behavior patterns were proposed to promote reliable communication and patient safety. Those suggestions are based on reported positive experiences, which could become mandatory instead of depending on personal commitment. Seniors could hold meetings and establish a trustworthy environment on a regular scheme. Furthermore, teaching mutual understanding for the challenges of the liaising professional groups is asked for in order to increase self-confidence and trust. In turn, uncertainties could be admitted and questions asked without feeling embarrassed.

#### 3.3.2. Ward/Clinic Level

At the ward level, incomplete or unclear documentation, inappropriate devices, and language barriers were named as the trigger for pAE. Additionally, a perceived steady increase in bureaucratic duties kept respondents from caring for patients. A rising number of patients and a high frequency of new colleagues are reported as further strains. Therefore, emergency trainings, language trainings for non-native speaking staff, training in precise communication, and hand-overs are proposed to reduce the risk for pAE.

#### 3.3.3. Health Care System Level

Concerning the health care system, understaffing was stressed as a huge burden. Overall, time constraints and a poor patient-staff ratio seemed to be accountable for many imperfect tasks. To address those shortcomings, improved staffing is requested. It would not only make everyday work easier, but also be perceived as the appreciation of professional engagements and strengthening of self-confidence. Furthermore, some midwives requested more professional recognition of their occupational skills, since they started academic training.

## 4. Discussion

To identify barriers and facilitators that impact safe communication in obstetrics from the subjective perspective of health care workers, we conducted qualitative interviews with a heterogeneous group of 20 physicians, midwives, and nurses. Therefore, we used a “work-as-done” [11] approach and started with the narratives of personal meanings and experiences in obstetrics. In the interviews, we balanced questions about supportive and resilient factors (regarding the safety II approach [19]) as well as adverse experiences, with the focus on the implementation of improvement and cohesion strategies.

### 4.1. Interprofessional Collaboration and Shared Responsibilities

Trustful communication is essential for safety-orientated care [24]. On a positive note, nearly all of our participants portrayed interprofessional collaboration as a valuable aspect. However, in line with prior research, data present how high job demands and unclear role perceptions lead to interprofessional conflicts [25]. Especially conflict pattern at the interface of professional boundaries became apparent. Physicians’ statements show that when urgent decisions are required, they value a biomedical point of view more than the bio-psycho-social approach which a midwife would take. While many physicians perceive their role as ”bad guys”, because they only take over if the birth becomes pathological, midwives sometimes may feel like they are overlooked due to hierarchical structures.

Unfortunately, nurses at the ward perceived their role as inferior, as they work beyond the “emergency zone”. Furthermore, hierarchy not only plays a crucial role between professions, but also in the cooperation between beginners and superiors within a profession. Professional boundaries may be overcome when experienced midwives train new residents which could enhance resilience in terms of safety II.

### 4.2. Speaking Up: Addressing Safety Concerns

However, effective safety culture requires corresponding knowledge, attitudes, competencies, and behavior from each individual working in health care [26]. Even interviewees who were actively pursuing a safety climate perceived hierarchy as a barrier to sufficient safety culture and speaking up. Again, part of this barrier is expressed as an interprofessional conflict. In that regard, some respondents report how their willingness to raise concerns about safety issues or asking potentially critical questions increases with social support. As speaking up appears to not depend on the professional status or expected social discomfort, these findings stress the importance of reliable working culture of mutual trust to improve patient safety [27,28]. Experiencing the need, as well as the possibility to speak up, is therefore, an important resource of health care workers, that needs to be implemented within safety II approaches.

### 4.3. Dealing with Uncertainty and Conflicts: Peer and Leadership Support

Doubts about appropriate approaches are reported as part of everyday work. While the request for peer support is described as normal, addressing doctors can be challenging due to perceived distance, untrusty personal relationships, or concerns about calling during a night shift. Furthermore, we found that conflicts between interdisciplinary groups are common—and remain mostly rather unsolved, more due to organizational hindrances than to personal ones. Our results concerning regular meetings to explicitly address cooperation patterns vary greatly. Debriefing and managing of conflicts seem to randomly depend on time constraints, the opportunity of meeting the staff involved shortly after an incident, and the superiors’ approach towards taking responsibility. In contrast, interviewees from all professional backgrounds and levels highly valued management support and debriefing, which clearly expressed superiors’ responsibility. That is, organizational commitment emerged as the main challenge. As previous research has shown, leadership culture coming from top-level governing to approachable management levels and thus front-line staff is essential for improved cooperation and reported as supportive [29,30].

### 4.4. Possible Trigger of pAEs and Improvement Suggestions

Even though the interviews focused on communication and insufficient communication was frequently identified as a determinant for pAEs, further aspects at various organizational levels were addressed by the study participants. In the following, we discuss the suggestions to improve patient safety corresponding to three levels of social systems, adapted to the institutions of the health care system [31,32]: team level, clinic level, and the health care system at large.

#### 4.4.1. Team Level

Our data present that communication failures occur on a daily basis, which plays a contributing factor in 70–80% of all patient safety incidents according to previous findings [5]. Against the background of reported supportive experiences, the willingness to actively contribute to improved patient safety plays an important role in strengthening resilience in health care [20]. Concordant with recent research, training approaches, especially emergency situation trainings including closed-loop communication were suggested to improve the allocation of tasks and check understanding during emergency situations [33]. Emergency simulation trainings are known to reduce patient morbidity [34]. In our study sites, they had been implemented before, though not for all employees. However, our interviewees still recalled miscommunication as a major patient safety concern which is explainable, as many factors influence speaking-up and mutual understanding including contextual and individual aspects, perceived efficacy, and interpersonal safety [35]. Consequently, underlying structures as hierarchy and interprofessional team performance need to be addressed, too [17]. This emphasizes the need for ongoing and repeated training, established and sustained by the hospital management identified according to the safety I approach [36].

Ineffective collaboration does not only affect the individual, but the resulting morbidity also causes high economic costs. In the British NHS alone, the estimated value of obstetric claims is £1.3 billion every year. This number justifies even expensive interventions targeting patient safety. Especially ongoing, systematic interventions have been found to increase patient safety in obstetrics, especially when targeting communication [37].

#### 4.4.2. Clinic level

On the hospital level, time constraints, shortage of staff, and insufficient documentation were reported as potential triggers causing pAEs. This is in line with prior research, that classifies technical conditions and organizational elements as particularly important for patient safety [38]. Furthermore, the rapidly changing staff was identified as a problem. Not only because their training consumes resources, but also as integrating into interprofessional teams is challenging. In addition, the shortage of staff causes tension, which in turn leads to unsolved conflicts within the team. A recently published study demonstrated how time pressure is directly related to a lower quality of care [39]. Since this finding reflects a “work-as-imagined” approach more than a safety-II-related “work-as-done” perspective, it is necessary to implement interventions based on positive care despite time constraints. On a higher level, the call for more personnel and lower patient–provider ratios still needs to be heard.

#### 4.4.3. Health Care System Level

Cost-cutting measures combined with insufficient quality assurance measures in the health care system have led to a more demanding patient-provider ratio in Germany [40,41]. Caring for several women at the same time is reported as an obstacle to patient safety, as multi-tasking reduces accuracy [42]. In our study, this is reflected as a major strain for all professional groups, who expressed the need for more personnel to spent longer time bouts with patients. Importantly, a recently published study presents that perceived good quality of care and higher job satisfaction seems to keep young physicians and nurses in Germany from leaving their professions [43]. Furthermore, the steady increase in bureaucratic duties reduces the time required for hands-on patients care. However, the current trend regarding digitalization tries to overcome this challenge and (in the notion of safety II) makes documentation more effective so work with similar patient-provider ratios becomes more feasible.

#### 4.4.4. Developing Patient Safety Approaches in Health Care Systems: Safety I to Safety II

In recent years, a new paradigm in patient safety has been advocated, shifting the perspective from what and why a task goes wrong (safety I) to focus on how often and why things go right (safety II). As indicated above, the safety II approach implies to see humans as “a resource necessary for system flexibility and resilience” [11] instead of evaluating the cause of human failure as in the safety I perspective. As we focused primarily on imperfect interprofessional communication skills, our study concept was initiated mainly on basis of the safety I approach. However, several encouraging examples came up during the interviews, e.g., learning from work experience regardless of professional hierarchy. Furthermore, most suggestions for improved communication, e.g., regular meetings, were taken from positive experiences. Otherwise, many hints were made concerning working conditions beyond team structure. That is, our interviewees strived for improved working conditions to facilitate improved communication and to expand on what goes right (safety II). In consequence, safety II is a promising approach, which needs to be investigated in further studies and applied to health care practice.

### 4.5. Legal Changes

In the near future, structural conflicts in obstetrics will be addressed by legal changes in Germany. Following the directive of the European Union, the training of midwives will be transferred from vocational schools to universities, accompanied by an increase of responsibilities of midwives. Although academization is widely supported, concerns about the implications are raised. While the legislator justifies the objective of the law as follows: “the academization also strengthens midwives in interprofessional cooperation. This is necessary with regard to their responsible work” [16,44], the National Association of Statutory Health Insurance Physicians (KBV) fears “that the unclear assignment of tasks by the professions will lead to even more accusations of treatment errors” [45].

These contrasting statements mirror the interprofessional conflict pattern we examined in our study. According to previous research, possible solutions need to follow different leverage points [36]. On a positive note, we found numerous suggestions for optimizing collaboration, again meeting the logic of safety II. For example, a new definition of teamwork and clear protocols that detail the responsibilities might activate resources and get closer to a thought-through clear decision and thus reduce uncertainty. Furthermore, examples are given in which professional experience is seen as an asset beyond professional status which may contribute to a safety II perspective acknowledging the positive aspects including climate, teamwork and appreciation of colleagues [20].

### 4.6. Limitations and Further Directions

Qualitative analysis is subjective by nature. We aimed to gain a deeper understanding of subjective perspectives on interprofessional collaboration. Although measures were undertaken to reduce interview bias, it cannot be completely excluded. As such, it is possible that findings may reflect the personal biases of the investigators. Along these lines, all participating physicians were specialized in obstetrics, i.e., as residents, specialists, or senior physicians and generalization to other professions remain open.

Furthermore, our findings present characteristics of German university hospitals where more women with critical pregnancies are admitted and therefore cannot claim generalizability [46]. In addition, most of our interviewees were female, and gender differences could not be investigated, or gender bias could not be controlled. Future studies are needed to validate the findings and include the safety II approach further. In this study, we have only presented the communication within the professional teams and not the patients’ essential view of their perception of communication. However, this needs to be covered in future research investigating this accordingly.

## 5. Conclusions

Our study provides insights into obstetric staff members’ opinion on interprofessional communication. Barriers and facilitators of safe communication in obstetrics can be manifold: Our findings indicate that communication, safety culture, individual effort and organizational management are closely entwined. Therefore, strategies on various levels of the health care system are necessary to improve patient safety in which communication training is a promising one.

Furthermore, legal changes could be accompanied by an in-depth analysis of the professional interfaces to integrate the responsibilities and competencies of all professions in protocols. In addition, the auspicious safety II approach, focusing on resilience in health care, should be investigated and supported further as it contributes to improve patient safety in real, mostly imperfect, working conditions. Starting with the perspective of professionals is just one cornerstone and further ones could follow up with focusing more on patients, and integrating the findings into existing tools [10] to learn from the subjective perspective of health care workers for organizational development and safety management.

## Figures and Tables

**Figure 1 ijerph-18-00915-f001:**
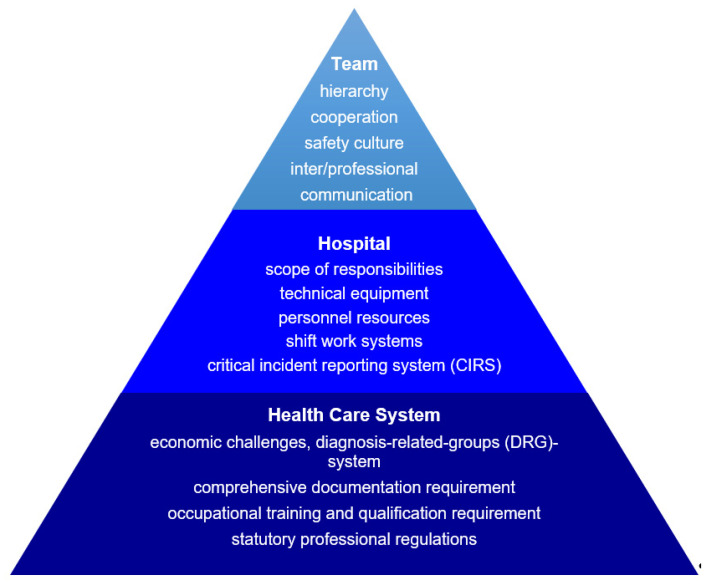
Structural levels affecting patient safety.

**Table 1 ijerph-18-00915-t001:** Semi-structured interview guide.

Introduction into Study Aim; Obtaining Informed Consent
How long have you been working in obstetrics? What was your reason to pursue this profession, this field of activity? Especially nice experiences? Unpleasant experiences? Main differences to other clinical fields?
From your point of view, how did the requirements change in the course of your professional life with regard to colleagues, superiors, and clinic management? Interdisciplinarity? Hierarchy/ support or hindrance? Role assignment/ allocation of responsibilities?
How did the demands of the mother/ patient and family members change in the course of your professional life?
To what extent does it happen that you get disappointed with superiors, colleagues, or patients? How do you cope with stressful situations at work? Talking to colleagues, to partners at home, silent suffering? Structured debriefing at work?
The core of the research study was about triggers regarding avoidable adverse events, which can lead to long-lasting consequences for mother and/ or newborn. To what extent are you familiar with such issues from everyday hospital life? In general, how do you deal with such challenges? (silent suffering, asking for collegial support) Blaming or supporting the staff involved?
Concerning safe communication: from your personal view, what does it require to ensure mutual understanding in everyday life? To what extent is this given in daily practice and when does it typically happen? What is missing among the colleagues to treat the women giving birth well and to care for the whole family? What would optimal communication look like for you?
Hierarchy gradient: There is evidence that greater hierarchy between the occupational groups is related to less willingness to point out possible errors when working together. How do you see this?
Autonomy: To what extent do you have the opportunity to shape the work processes, i.e., to make suggestions for improvements?
Wishes: Which improvements or changes in the daily work routine could you think of to improve communication?
Which other comments or questions do you have? Thanks for your openness and your time!

**Table 2 ijerph-18-00915-t002:** Demographic characteristics (no numbers for gender among the profession are displayed due to data protection requirements).

	Midwives N = 7	Physicians N = 8	Nurses N = 5	∑ = 20
Female	*	*	*	18
Male	*	*	*	2
Migrant (first generation)	2	1	2	5
Occupational age				
<5 years	2	5		7
5–15 years	4	1	1	6
>15 years	1	2	4	7
Professional Position				
Senior, Superior	1	3	1	5
In Training, Resident	3	4	0	7

Note. * Gender was not tracked per job group to ensure anonymity and data protection of the participants.

**Table 3 ijerph-18-00915-t003:** Task sharing.

Task Sharing in Everyday Life between All Professional Groups			
Midwives	Physicians		Nurses
This side, which had doubts about the decision, mostly the doctors, prevailed with their decision, because they simply, so to speak, are above it in the clinical hierarchy (P20Midwife). I have the impression that they (the nurses) sometimes feel very excluded, because if they need anything at the ward, then it takes a while until the help called comes but when the delivery room calls, they react immediately, but maybe a birth with the bleeding is a bigger emergency than having to give an IV access (P5Midwife).	It often means that we have to terminate the birth relatively quickly because the woman has been under labor for a very long time and we simply have risks in mind that the midwife has not yet considered or that she judges differently from the dynamics of birth (P1Physician). It happens quite often, that we are perceived as bad guys because in principle, we get into play when the birth becomes pathological. And the midwife doesn’t even want to be mean, she has her own view on the path, she thinks, we pathologize it (TN9Physician).		In the first place there are the doctors, then the midwives, of course, and then, we are at the bottom, exactly (P17Nurse). … you have to sort things out at the ward round or by explaining any diagnosis. This is so undetermined for the women, where they only think about it afterwards and need to talk again (P8Nurse).
**Task shifts between midwives and physicians**			
**Midwives**		**Physicians**	
We usually don’t involve the doctors very much and I would honestly say that they do not mind; if they only have to appear to the birth, the child is born, they congratulate and then continue working. I think they also enjoy it and that’s the same from our side, if we can just call them to the birth and then maybe they need to do some stitches afterwards and then everything is good and they can do their business again (P5Midwife).		So, I would also like to be present at the births much longer, but in the end, it is that I introduce myself briefly and if there are no problems during the birth, then you come at the end once briefly; (…) and it would be naturally nicer if you could comfort the women a little (…) but I don’t think it’s really feasible with the time available (P4Physician). Yes, and also the “empathic breathing”, so to speak, and calming the patient during birth, that is not what I studied medicine for, either. Well, I don’t want to take that over from the midwives, not at all (P9Physician).	
**Collaboration between midwives and residents**			
**Midwifes**		**Physicians (residents)**	
I think the residents are, well, at the beginning very needy and attach great importance to what the midwife says and are happy about ideas, opinions and advice (P3Midwife). But there are colleagues on the doctors’ side who say, ‘No, you carry on’, they are there, but if an emergency situation arises, I simply have more experience than doctors, then they also say ‘yes, you are in charge, what should I do, what we might have forgotten?’, so that’s partly a very cooperative teamwork. It’s nice when you are also appreciated for what you can do (P6Midwife)		Professional experience is, I think, a very important point, no matter which professional group it is. If a midwife has been in the job for forty years now, I can have studied as much as I want, if I am only one year in the job, she will certainly have more practical experience than I have (P15Physician). The midwives actually work much more independently and competently, I believe that it is much easier for me as a beginner (…) If you have an experienced midwife, you can rely on her and she also teaches me a lot, because she has much more experience, which is precious. I am glad to have them (P4Physician).	

**Table 4 ijerph-18-00915-t004:** Managing conflicts and speaking up from the different professional perspectives.

Speaking Up: Addressing Safety Concerns			
Midwives		Physicians	
I think that, at the senior physician’s, the responsibility is almost transferred, whereby I think we still have to share the responsibility, because if something goes extremely wrong and I have recognized that, but have not expressed it, then not legally, but emotionally, so then you’ll think if maybe you would have expressed your opinion and could have taken it into another direction. Well, I think with the *young doctors* it’s more difficult in terms of responsibility because maybe I have more experience and I think that you have to say this. And to the senior physicians, it’s really a big obstacle to say: I see it differently and what do you think about doing it this way? It’s not always that easy, it depends on the type of doctor, different doctors, of course (TN3Midwife). Well, the doctors have studied for a long time and are senior physicians and have perhaps already done their own research, so in this situation, I find it quite difficult to speak up and to intervene with my opposing assessment, mainly because I don’t know whether it really helps, or whether it finally complicates things (P20Midwife).		But when I see problems, I have to keep them in my mind until the end until I have actually solved them in the end. Yes, and everyone can say, ‘you see it wrongly, we do it our way, but then I would like to say: ok, then do it without me, until the end. Yes, but when I am called in the end (…) then it’s my problem (P9Physician). I think that sometimes the younger midwives don’t dare to give their point of view, which is probably right, because they think the doctor is older, more experienced or has to make the decision right now and we sometimes don’t oversee it and maybe we just don’t question. And I don’t know much more than the midwife. (…) And we young residents feel sometimes more restrained with our opinion and just don’t communicate. It’s just in your head and you don’t say it out loud (P18Physician). Of course, there are situations where you have the feeling that if a very experienced midwife now suggests something, that you might not be in a position to disapprove (P15Physician).	
**Dealing with uncertainty and conflicts: Peer and leadership support**			
**Midwives**	**Physicians**		**Nurses**
I ask my colleagues. Or I ask the junior colleagues because they know things better from theory (P6Midwife). In the early or late shift, to call a doctor is no problem (...). If it is 4 a.m. and I know he is sleeping and I am just a little bit unsure, I feel more uncomfortable calling him. Well, it is shift-dependent, it is a bit doctor-dependent and of course related to the different cases (P5Midwife). It developed over time that we dare to approach the leadership. I know that the residents do not yet dare to go to the management if there are problems, and we midwives are already a bit tougher because we (…) cannot always assert ourselves on our own (P3Midwife).	It is a great advantage that one can ask a lot, in any case, (...) especially as a beginner, i the first job, the first experience you gain; if you somehow get the feeling that you can’t ask everyone, I imagine that this would be very demanding. So, I was glad that it was always possible (P15Physician). (...) often after the shift, you discuss it again; (…) often during the situation, unfortunately, this is not possible, because the telephone rings or something else comes in between (…). With the midwives, it is sometimes more difficult because when they have an earlier shift change than we have, you don’t see them until a few days later, then the debriefing is not so immediately possible (P4Physician).		Well, it’s very important that I can say that something happened to me and that I do not have to be afraid of the hierarchy. (…) Everyone has different abilities. Well, I don’t know anything. There are people who know some things much better than I do. And they should be able to apply it accordingly (P8Nurse). And sometimes you don’t even dare to ask something because you see that it is not welcomed (P16Nurse).
When conflicts arise, there is always the possibility for a conversation, that’s what we are looking for. Thus, if I see such situations, then in any case the conversation is sought (…). If several individuals are affected, then a case is of course also discussed in the team (P7Midwife).	For me it’s clearly a responsibility depending on the leadership position; simply, one implements and exemplifies it from top to bottom. I think the more one just does it, the more it will be continued by the required groups of people (…) doctors with midwives interdisciplinary (P1Physician).		We also had some difficulties; now we have a working group with individuals from the ward and the delivery room. We discuss with our team leader what we did not like, so things work a bit better (P11Nurse).

## Data Availability

Original data are not available for data protection reasons.

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
