# Peer review of "Barriers and Facilitators of Safe Communication in Obstetrics: Results from Qualitative Interviews with Physicians, Midwives and Nurses"

_ijerph, 2021, doi:10.3390/ijerph18030915_

Round 1

Reviewer 1 Report

Thank you for the opportunity to review your manuscript.  I only have few minor comments. Overall, your manuscript focuses on qualitative aspect of mixed-methods research project, and therefore, is limited in scope of qualitative interviews. 

  1. You have enlisted 20 interviewees. Could you specify how many interview requests were sent? Sampling Bias is always of concern and should be addressed.

  2. Please update Table 2, so it is easier to follow as a reader. <5, 5-15, +15 of Occupational Age should be indented to let readers know that those fall under the different categories of occupational age. Same with Professional Position.

  3. In Table 2, are # of Female and # of Male providers intentionally left blank? 

  4. Under 2.1 study design, authors mention that field notes were taken to document impressions of atmosphere, nonverbal communication...    Yet there are no mention of impression of these on-verbal communications by interviewees.  Either elaborate further or state non-difference.

  5. Please clarify the roles of physicians. As your manuscript states, there are multiple specialty of physicians playing a role in LnD units (Obstetric, Pediatric and anesthesiology). 

  6. Under limitation: Would consider mentioning that most of your interviewees were female. Gender difference and gender bias, especially if midlevel providers (RNs and midwives) are different gender than physician providers. 

Author Response

Thank you for the opportunity to review your manuscript. I only have few minor comments. Overall, your manuscript focuses on qualitative aspect of mixed-methods research project, and therefore, is limited in scope of qualitative interviews.

AUTHORS: Thank you for your helpful comments, which we have all taken into account in the revision, as shown below.

1. You have enlisted 20 interviewees. Could you specify how many interview requests were sent? Sampling Bias is always of concern and should be addressed.

AUTHORS: We have explained the sampling strategy in more detail in the text (paragraph 2.1 line 113). There have been no refusals of potential interviewees to participate in the study. We proceeded in the following steps: In the first step, the composition of the sample was planned in terms of occupation and work experience. In the second step, the local research staff in the clinic asked potential respondents whether they were willing to participate in the study if they matched the criteria of the research design. In the third step, the interviews were carried out. It is important to note that the interviewer adjusted the time schedule of the interviews to the requests of respondents. This helped to avoid refusals together with the high commitment of the obstetric ward to the study.
In addition to the complemented information (see revised manuscript highlighting the changes), we also added a general reference highlighting the importance of sampling as well as avoiding sampling biases. It also helps outlining the line of thought and methodological approach we followed when conducting the study:
Schreier, M. (2018). Sampling and generalization. In U. Flick (Ed.), The Sage handbook of qualitative data collection (pp. 84-98). London: SAGE: Publication.
In lines 151ff we now report “Stratified purposive sampling [19] was accomplished although few men and no nurses with fewer than 5 years in the job (occupational age) could be recruited. There have been no refusals of potential interviewees to participate in the study.”

2. Please update Table 2, so it is easier to follow as a reader. <5, 5-15, +15 of Occupational Age should be indented to let readers know that those fall under the different categories of occupational age. Same with Professional Position.

AUTHORS: We improved the table accordingly (line 148).

3. In Table 2, are # of Female and # of Male providers intentionally left blank?

AUTHORS: Yes. As we mentioned on the top of the table “no numbers for gender among the profession are displayed due to data protection.” The intention is to protect personal information. To give an insight into the background: the participation group is selected from a known population of about 160 employees from two clinics who are easily to be identified. Contrary to expectations, the two male participants are midwives. As the number of male midwives in Germany is very low, we decided to mask gender in both the quantitative and qualitative data throughout the project.
Accordingly, we added to Table 2:
“Note. *Gender was not tracked per job group to ensure anonymity and data protection of the participants.” (lines 149f).

4. Under 2.1 study design, authors mention that field notes were taken to document impressions of atmosphere, nonverbal communication… Yet there are no mention of impression of these on-verbal communications by interviewees. Either elaborate further or state non-difference.

AUTHORS: We have used field notes in the study as a common tool of qualitative research in order to improve the interpretation of the qualitative data such as uncertainty, discomfort, but also the feeling of reluctance or unwillingness or lack of time. All this can influence the quality of the results, e.g. because the interviewer asks questions too cautiously in the beginning or because a person only starts to open-up in the course of the interview. Therefore, field notes are used for the later interpretation also as an orientation for the researchers who are evaluating but did not conduct the interview. The results are directly included in the evaluation and assignment of the interview sequences. Sometimes the answers given by professional interviewees are rather affirmative at the beginning: "We have no problems here" and only in a later part they are reported in more detail. Hand movements or grimaces that are not recorded on the tape are also noted. To describe the meanings of the field notes, an explanatory sentence was added: “…and used for better understanding and interpreting of the orally recorded interviews (lines 124ff).

5. Please clarify the roles of physicians. As your manuscript states, there are multiple specialty of physicians playing a role in LnD units (Obstetric, Pediatric and anesthesiology).

AUTHORS: Thank you for this important note. We have clarified in the manuscript that all participating physicians are specialized in obstetrics, i.e. as residents, specialists or senior physicians (please see the manuscript high-lightening the changes). This information is also added to the limitation and it now reads “Along these lines, all participating physicians were specialized in obstetrics, i.e. as residents, specialists or senior physicians and generalization to other professions remains open” (lines 449ff).

6. Under limitation: Would consider mentioning that most of your interviewees were female. Gender difference and gender bias, especially if midlevel providers (RNs and midwives) are different gender than physician providers.

AUTHORS: Thank you for this remark. As disclosed in comment to question 2, all physicians were female, but two midwives were male. We agree that gender bias is a very important issue in medicine. However, we cannot address this issue in our specific sample. Therefore, we now state “Also, most of our interviewees were female, and gender difference could not be investigated or gender bias could not be controlled. Future studies are needed to validate the findings…” (lines 454).

Reviewer 2 Report

Authors conducted Interview-Based Study to clarify the barriers and facilitators of safe communication in obstetrics. This is important issue to reduce the preventable adverse evets. This manuscript demonstrate abundant information, however, it is difficult to interpret what is the novel and significant findings. As demonstrated in this manuscpript, there are several factors affecting safe communication, however, these factors are actually assumable even without this interview based study. Readers are much more intersted in ranking of each potential factor.

Table 3 is the result and most importat part of this study, however, it is difficult to understand what authors intend to demonstrate from this table, because this table is too wordy. Authors should reconsider the expressing way of these table. Among candidate factors, authors should demonstrate major and minor factors from this interview-based study. It is better to demonstrate the data that can be understood visually, for example, showing rating or percentage of how many interviewee claim each opinion.

Author Response

AUTHORS: Thank you for your helpful comments, which we have all taken into account in the revision, as shown below.

Authors conducted Interview-Based Study to clarify the barriers and facilitators of safe communication in obstetrics. This is important issue to reduce the preventable adverse evets. This manuscript demonstrate abundant information, however, it is difficult to interpret what is the novel and significant findings. As demonstrated in this manuscript, there are several factors affecting safe communication, however, these factors are actually assumable even without this interview based study. Readers are much more interested in ranking of each potential factor.

AUTHORS: Thank you for your comment. Indeed, research about the impacts on safe communication was done in many countries. However, to the best of our knowledge, this is the first study conducted in Germany and we note that with the following sentence: “As there has been previous research, to the best of our knowledge, this is the first study conducted in Germany” (lines 90ff).
Please note that the health care system in general and the system of obstetrics displays in Germany many specific features which makes this a special case relative to other countries. Most of the ‘non-physician’ health care workers, as they are called in Germany, are dependent from doctor’s advice which makes cooperation different compared to other countries. Therefore, there is a huge discussion about “delegation and substitution” of certain tasks from physicians to ‘non-physician’ health care workers in Germany. Therefore, the perception of midwifes within the birthing process stresses the importance of professional self-determination and task shifting. As mentioned in the discussion section “In the near future, structural conflicts in obstetrics will be addressed by legal chances In Germany. Following the directive of the European Union, the training of midwives will be transferred from vocational schools to universities, accompanied with increased responsibilitis of midwives of midwives…” (lines 431ff). Against this background, structural professional conflicts will need to be addressed sustainably by the legislator and the professional associations and we hope to cover part of it.

Table 3 is the result and most important part of this study, however, it is difficult to understand what authors intend to demonstrate from this table, because this table is too wordy. Authors should reconsider the expressing way of these table. Among candidate factors, authors should demonstrate major and minor factors from this interview-based study. It is better to demonstrate the data that can be understood visually, for example, showing rating or percentage of how many interviewee claim each opinion.

AUTHORS: Following your suggestion to rank the importance of the results, we revised the structure of the tables to make them more readable, see Table 3.1a, Table 3.1b and Table 3.2.
We divided table 3.1 into two parts and moved table 3.2 to the appendix. We clarified in the discussion section that the first part (new table 3.1a) represents the daily professional life from the respective views and that the second part (new table 3.1b) provides insights into the conflicts which hinder good communication. These are the most important results concerning our research question. The old table 3.2 has been moved to the appendix since the most important message, factors which have an external influence on the communication, are mapped in figure 1.
Furthermore, we have thoroughly considered how a weighting of the answers could look like. However, as the qualitative approach strives to reveal patterns of subjective perceptions, the interview structure followed the narratives of the respondents. Therefore, a rating of topics might bias the presentation of the results and should be done in a different, e.g. quantitative, research setting. Accordingly, we now state in the discussion section “Future studies are needed to validate the findings and include the Safety-II approach. In this study we have only presented the communication within the professional teams and not the patients' essential view of their perception of communication. However, this needs to be covered in a future research investigating this further accordingly” (lines 455ff).

Reviewer 3 Report

Safe communication has a great positive impact on patient safety and the ability to prevent adverse events (AEs). To create awareness and knowledge about the human factors that influence patient safety, this study aimed to identify barriers and facilitators of safe communication in obstetrics, based on the subjective perspectives of physicians, midwives and nurses of two German hospitals. Barriers and facilitators can be assigned to three levels of analysis: the team, the clinic and the health care system. Time-constraints and understaffing were mentioned as barriers for safe communication. There were a few important improvements mentioned, namely: interprofessional meetings to improve collaboration between physicians, midwives and nurses, and training in precise communication to reduce the risks for AEs. Time-constraints and understaffing were mentioned as barriers for safe communication.

Major comments:

  • I deeply miss the patient perspective in this study. Obviously there is also room for studies of the professional perspective only, but these should be complemented by studies from the patient perspective. That should at least be acknowledged in the discussion.
  • Also, this study seems to equal quality with the absense of mistakes and AE. While that is certainly important, this is nowadays generally refered to as safety-1 type of thinking. The more modern type of thinking is safety-2, where the emphasis is more on how to enhance good quality than on the prevention of AE. The authors should at least refer to this line of thought.
  • The study is broadly speaking performed in a satisfactory way: the qualitative work is done nicely (with some suggestions, see below) and described well. The introduction to the study however is really weak: there are many sentences (also in the discussion: line 242) in which the goal/ aim of the study is rephrased which is confusing. It should be noted that safe communication does not equal communication about safety nor does it equal looking at communication through the lens of safety. The authors mix all these issues, which really should be remedied. Currently the real focus is on perceived barriers and facilitators. Please state so throughout the article when describing the aim.
  • the rationale for the study is lacking. What does this study add? which kowledge gap do you want to fill? the sentence in line 48-49 is not at all convincing. The reference given dates from the year 2000 and the article itself quotes many refs that show that more knowledge is available ( eg 12, 14, 15, 18, 9, 20). So the introduction really needs to be rewritten in such a way that more justice is done to what is known already. That will also help in establishing which specific gap this study can fill.
  • The study shows that communication, a safety culture, individual efforts and management are important for the safe communication in obstetrics. In addition, it provides two recommendations that can be implemented to increase safe communication: communication training and in-depth analysis of professional interfaces. At this point these conclusions seem rather obvious and not surprising. Didn't we know this already?
  • The interviews, coding, and analysis is mainly done by one of the authors (MS). Was there a quality check of the codes/ analysis besides the subsequent discussion with the research group and participants? Did a second author check the coding? this would be necessary.

Specific comments:

  • line 104: remove “collection”
  • Structure of the results and discussion. In the discussion, another objective/ aim is mentioned. Plus, in the discussion, some of the results are merely repeated. As a reader you are presented with many different translations of the results.
  • Move table 3.1 and 3.1 to appendix?

Author Response

AUTHORS: Thank you for your helpful comments, which we have all taken into account in the revision, as shown below.

Major comments:

• I deeply miss the patient perspective in this study. Obviously there is also room for studies of the professional perspective only, but these should be complemented by studies from the patient perspective. That should at least be acknowledged in the discussion.
AUTHORS: Thank you for emphasizing this important point. We share the reviewer’s view that the patients’ perspective is highly relevant for a better understanding of patient safety. However, in this publication we only present the communication within the professional teams. Nevertheless, we explicitly include this point as we focus on subjective theories also relating to the patient perspective. According, we also included this in the intro and the sentence now reads “While the patient perspective is important, the professional perspective with subjective theories about patient safely behavior and proximal determinants should be focused on first” (lines 52ff).
Though, in a following publication, we will present further results of our TeamBaby research project focusing on the patients’ perspective. To clarify this, we included this limitation and further information in the discussion section. It now reads “Future studies are needed to validate the findings and include the Safety-II approach. In this study we have only presented the communication within the professional teams and not the patients' essential view of their perception of communication. However, this needs to be covered in a future research investigating this further accordingly” (lines 457ff).

Also, this study seems to equal quality with the absense of mistakes and AE. While that is certainly important, this is nowadays generally referred to as safety-1 type of thinking. The more modern type of thinking is safety-2, where the emphasis is more on how to enhance good quality than on the prevention of AE. The authors should at least refer to this line of thought.

AUTHORS: We also agree with this point of the reviewer and have revised the passage in the introduction accordingly: “However, to implement strategies to improve safety, safety management needs not only to ask why things go wrong, but also to acknowledge that most of the care works well, as clinicians adjust their work to real life conditions. In addition, to avoid a pAE, it should be looked at how and why it usually works well” (lines 49ff).

• The study is broadly speaking performed in a satisfactory way: the qualitative work is done nicely (with some suggestions, see below) and described well. The introduction to the study however is really weak: there are many sentences (also in the discussion: line 242) in which the goal/ aim of the study is rephrased which is confusing. It should be noted that safe communication does not equal communication about safety nor does it equal looking at communication through the lens of safety. The authors mix all these issues, which really should be remedied. Currently the real focus is on perceived barriers and facilitators. Please state so throughout the article when describing the aim.

AUTHORS: Thank you for this suggestion. We revised the introduction and clarified the goal of the study taking this suggestion of the reviewer on board.

• the rationale for the study is lacking. What does this study add? which kowledge gap do you want to fill? the sentence in line 48-49 is not at all convincing. The reference given dates from the year 2000 and the article itself quotes many refs that show that more knowledge is available ( eg 12, 14, 15, 18, 9, 20). So the introduction really needs to be rewritten in such a way that more justice is done to what is known already. That will also help in establishing which specific gap this study can fill.

AUTHORS: Thank you for these detailed notes. We revised the introduction and outlined the rationale of the study as well as an elaboration of the value of its contribution to the state of knowledge accordingly, inter alia by following the suggestion of stressing the rationale of implementing safety cultures.
Accordingly, we added significant references (updated:)
8. Iflaifel, M., et al., Resilient Health Care: a systematic review of conceptualisations, study methods and factors that develop resilience. BMC Health Serv Res, 2020. 20(1): p. 324.
17. Braithwaite, J., R.L. Wears, and E. Hollnagel, Resilient health care: turning patient safety on its head. Int J Qual Health Care, 2015. 27(5): p. 418-20.
21. Hollnagel, E., R.L. Lears, and J. Braithwaite, From Safety-I to Safety-II: A White Paper. 2015.

• The study shows that communication, a safety culture, individual efforts and management are important for the safe communication in obstetrics. In addition, it provides two recommendations that can be implemented to increase safe communication: communication training and in-depth analysis of professional interfaces. At this point these conclusions seem rather obvious and not surprising. Didn't we know this already?

AUTHORS: Indeed, many results of this study are well-established in the international literature on other countries. However, to the best of our knowledge, this is the first study conducted in Germany, where the setting in obstetrics has many specific features. In particular, the so-called ‘non-physician’ health care workers are dependent from doctor’s advice there, which distinguishes the prerequisites for communication in obstetrics from that in other countries. Therefore, there is a huge discussion about “delegation and substitution” of certain tasks from physicians to ‘non-physicians’ health care workers in Germany, not only concerning professional responsibilities and standards, but also in terms of budgeting. Therefore, the perception of midwifes within the birthing process stresses the importance of professional self-determination and task shifting. As mentioned in the discussion section, legal changes will soon lead to increased responsibilities of midwives, following a directive of the European Union. Against this background, structural professional conflicts will need to be addressed sustainably by the legislator and the professional associations. We hope to contribute with our study to provide new insights on the need of supporting measures. Accordingly, we state in the discussion:
“In the near future, structural conflicts in obstetrics will (probably) be addressed by legal changes in Germany. Following the directive of the European Union, the training of midwives will be transferred from vocational schools to universities, accompanied with increased responsibilities of midwives. Although academization is widely supported, concerns about the implications are raised. While the legislator justifies the objective of the law as follows: ‘the academization also strengthens midwives in interprofessional cooperation. This is necessary with regard to their responsible work’ [14, 46] the National Association of Statutory Health Insurance Physicians (KBV) fears ‘that the unclear assignment of tasks by the professions will lead to even more accusations of treatment errors’ [47]. These contrasting statements mirror the interprofessional conflict pattern we examined in our study. According to previous research, possible solutions need to follow different leverage points [36]. On a positive note, we found numerous suggestions for optimizing collaboration. For example, a new definition of teamwork and clear protocols that detail the responsibilities might activate resources and get closer to a thought-through clear decision and thus reduce uncertainty. Furthermore, examples are given in which professional experience is seen as an asset beyond professional status which may contribute to a Safety-II perspective [17] acknowledging the positive aspects including climate, team work and appreciation of colleagues“ (lines 429ff).

• The interviews, coding, and analysis is mainly done by one of the authors (MS). Was there a quality check of the codes/ analysis besides the subsequent discussion with the research group and participants? Did a second author check the coding? this would be necessary.

AUTHORS: Thank you for this question. We complemented the result section to address this issue. Two other researchers, JD and FH, checked and revised the codings independently and separately from each other. The final assignment was made based on a consensus of all three researchers (MS, JD, FH). (Accordingly, the method section now reads: “Based on this structure, attitudes, perceptions and explanation patterns were compared and contrasted between and within occupational groups. Then, JD and FH checked and revised the codings and assigned them, independently and separately, to the categories. Finally, main categories were built to answer the research question. Consensus was sought (MS, JD, FH) until agreement was achieved” lines 157ff).

Specific comments:

• line 104: remove “collection”
AUTHORS: Done.

Structure of the results and discussion. In the discussion, another objective/ aim is mentioned. Plus, in the discussion, some of the results are merely repeated. As a reader you are presented with many different translations of the results.

AUTHORS: We revised the section and complemented by discussing our results referring to the safety-II concept. (see the revised manuscript highlighting the changes)

• Move table 3.1 and 3.2 to appendix?

AUTHORS: We kept table 3.1 but displayed it in two parts (see our reply to reviewer 2). Also, we optimized table 3.2 following the suggestion of the reviewer and removed it to the appendix.

Round 2

Reviewer 2 Report

Authors address important issue regarding barriers and facilitators of safe communication in obstetrics.

However, manuscript is too wordy, so authors should further limit the number of main points.

The purpose of this study is to investigate how safe communication can be achieved in obstetrics setting. So, contents other than this point should be excluded. For example, I think description about the job motivation is not necessary, because it dosen't develop the matter of safe communication in this manuscript.

Discussion is also too wordy and should be further limit the number of main points. In discussion, author should compare the previous reports and discuss the novel and significant findings of this study.

Figure 1 is great to demonstrate the important issues of this study visually and clearly.

Author Response

Review 2

Authors address important issue regarding barriers and facilitators of safe communication in obstetrics.

However, manuscript is too wordy, so authors should further limit the number of main points.

The purpose of this study is to investigate how safe communication can be achieved in obstetrics setting. So, contents other than this point should be excluded. For example, I think description about the job motivation is not necessary, because it dosen't develop the matter of safe communication in this manuscript.

Authors:

Thank you for this feedback. We followed the helpful suggestion by the reviewer and removed the results concerning job motivation from the table (line 238f). Furthermore, we deleted all paragraphs on professional motivation and paragraphs to focus on the investigation of safe communication. Additionally, we have proofread and revised our manuscript to make it less wordy also with proofreading help, whom we now acknowledge accordingly: “We also appreciate the support of Jing Liu for supporting us with language editing and proofreading” (line 478).

Discussion is also too wordy and should be further limit the number of main points. In discussion, author should compare the previous reports and discuss the novel and significant findings of this study.

Authors:

Thank you for this feedback. To focus on the main points, we deleted paragraphs about job motivation, summarized results about task shifting and deleted summaries about the contents of the tables. Again, we asked a new proofreader for help to make the discussion less wordy and acknowledge her accordingly: “We also appreciate the support of Jing Liu for supporting us with language editing and proofreading” (line 478).

Figure 1 is great to demonstrate the important issues of this study visually and clearly.

Authors:

Thank you!

Reviewer 3 Report

I would like to thank the authors for their reply. In many ways the manuscript is improved. However, my major concerns still are:

  1. De rationale for the study is still pretty weak. It seems this is relevant for Germany only. So why write about it in an international journal? What can others learn? Is there really a knowledge gap that is filled? if so, which?
  2. First paragraph of the discussion (lines 316-321) is the best description of the aim and method. However, it includes an introduction to a new approach: work-as-done. Introduce work-as-done in introduction (since this approach has had “consequences both for how adverse events were studied and for how safety could be improved”), refer to the approach in the discussion
  3. I am glad the authors agree with me about the importance of the patient perspective and of safety 2. Right now the article seems to suggest that the presented work is also relevant from a safety2 perspective, and I think that this is not convincing. This project is an example of safety 1 type of thinking, which isn't wrong, but should be supplemented by safety 2. To present this project as saftey 2 is somewhat misleading

Some specific remarks:

  1. l.52 While the patient perspective is important, the professional perspective with subjective theories about patient safety behavior and proximal determinants should be focused on first. why SHOULD??
  2. What is the message of lines 55-61. Message is unclear. Skip para?
  3. L 57. Though, health care systems are complex organizations, whose outcomes are determined by financing structures, occupational training, workload and professional regulation [11]. replace “though” by Nonetheless?
  4. l. 59: still no clear explanation of what safe communication entails or why the focus on communication is essential.
  5. l 95: comma between “health care” and “by means”
  6. I don’t get the change from special feature of obstetrics to job motivation in obstetrics. It does not seem to be the topic the participants are referring to.
  7. Summaries of the tables 3.1 and 3.2 are mainly about the differences in perspectives, rather than a summary of the content of these perspectives. What is the point in telling everything is different? We can read that ourselves.
  8. Note on p. 11 line 285 unnecessary

Author Response

I would like to thank the authors for their reply. In many ways the manuscript is improved. However, my major concerns still are:

  1. The rationale for the study is still pretty weak. It seems this is relevant for Germany only. So why write about it in an international journal? What can others learn? Is there really a knowledge gap that is filled? if so, which?

Authors:

Thank you for indicating an important shortcoming in the introduction. Accordingly, we clarified our study aim, to present an in-depth-analysis of collaborating staff on which improvement strategies can be built in a further step. In our opinion, barriers and facilitators of safe communication can accordingly be found in all university hospitals in developed countries. A more in-depth understanding is therefore relevant in more countries than Germany. In the text, we deleted the focus on Germany to avoid misunderstandings in this regard:

The abstract is now worded:

“…we aimed to identify barriers and facilitators that impact safe communication in obstetrics from the subjective perspective of health care workers”  (line 19) and the  manuscript now reads:

 “While the patient perspective is a crucial indicator of quality problems and improvement in health care, health care providers are working more constantly in the setting and are held responsible if pAEs occur. Since they are the experts for their ‘work as done’, improvement strategies in the clinical setting, focusing on the health care providers are needed. This study was conducted as part of the “TeamBaby – safe, digitally supported communication in obstetrics” project which aims to avoid pAEs by implementing safe communication in different ways. Therefore, we started with subjective theories from the professional perspective of proximal determinants of patient safety and adverse events to base future improvement measures on the results. Our approach allows us to study the patient safety from a so-called “safety I” perspective. This perspective is to identify, understand and overcome the sources of failure. In contrast, the “safety II” approach aims to reach a more positive perspective with understanding and improving how processes lead to good care in everyday clinical work [13]….” (line 62f)

as well as:

“…the aim of the study was to identify barriers and facilitators that impact safe communication in obstetrics from the subjective perspective of health care workers by means of qualitative interviews with a heterogeneous group of 20 physicians, midwives and nurses” (line 109f).

We hope that the rationale for the study is now more transparent and how we aim to fill the knowledge gap.

First paragraph of the discussion (lines 316-321) is the best description of the aim and method. However, it includes an introduction to a new approach: work-as-done. Introduce work-as-done in introduction (since this approach has had “consequences both for how adverse events were studied and for how safety could be improved”), refer to the approach in the discussion.

Authors:

Thanks, we have revised our manuscript accordingly: We now introduce the work-as-done approach into the introduction section (line 56f) as suggested:

 “In addition, to avoid pAE, it could be looked at how and why care is delivered well in most cases. This ‘work-as done’-approach focuses on tasks under given working conditions instead of ‘work-as-imagined’, which tends to ignore obstacles staff has to cope with in everyday life” [11]. The approach covers both, how adverse events were triggered and how safety could be improved based on quality-based knowledge.” In the discussion section we referred to this approach (line 408f).

I am glad the authors agree with me about the importance of the patient perspective and of safety 2. Right now the article seems to suggest that the presented work is also relevant from a safety 2 perspective, and I think that this is not convincing. This project is an example of safety 1 type of thinking, which isn't wrong, but should be supplemented by safety 2. To present this project as safety 2 is somewhat misleading.

Authors:

Thank you, we very much appreciate the safety 2 approach and now expanded on this in the introduction with:

 “In addition, to avoid pAE, it could be look at how and why care is delivered well in most cases. This ‘work-as done’-approach focuses on tasks under given working conditions instead of ‘work-as-imagined, which tends to ignore obstacles staff has to cope with in everyday life” (line 56f).

However, we did not want to indicate falsely that our study was based on the safety 2 approach. We therefore edited the introduction and discussion to show that the main approach was safety 1 while including some elements from safety 2. Although we focused on resources and facilitators of positive communication, our results and discussion present negative aspects of healthcare that need to be addressed in order to avoid pAEs (which clearly reflects safety 1).

In the discussion, we clarified that our study concept bases on the safety 1 approach, but covers some elements of safety 2, as respondents reported encountering examples of good communication and backed most improvement suggestions on positive experiences. It now reads:

“As we focused primarily on imperfect interprofessional communication skills, our study concept was initiated mainly on basis of the safety I approach. However, several encouraging examples came up during the interviews, e.g. learning from work experience regardless of professional hierarchy. Furthermore, most suggestions for improved communication, e.g. regular meetings, were taken from positive perspectives. Otherwise, many hints were made concerning working conditions beyond team structure” (line 412f).

Some specific remarks:

  1. l.52 While the patient perspective is important, the professional perspective with subjective theories about patient safety behavior and proximal determinants should be focused on first. why SHOULD??

Authors:

We apologize for this language error and have now replaced all instances with could. Furthermore, we clarify that we started with the professional perspective of proximal determinants because we planned to develop improvement measures on those results.

What is the message of lines 55-61. Message is unclear. Skip para?

Authors:

With the numbers, we strive to underline the importance of improvement in patient safety.

L 57. Though, health care systems are complex organizations, whose outcomes are determined by financing structures, occupational training, workload and professional regulation [11]. replace “though” by Nonetheless?

Authors:

Thank you, we have changed though to nonetheless (now line 60).

  1. 59: still no clear explanation of what safe communication entails or why the focus on communication is essential.

Authors:

We described in detail that communication encompasses important skills leading to patient safety. It now reads:

“As communication encompasses the essential ability not only to pass clear information, but also to reconsider and discuss challenging decisions in treatment on team level, we aimed this study in a field associated with high risk for patients to understand case of communication and resilience in health care systems “ (line 74f).

l 95: comma between “health care” and “by means”

Authors:

Thank you, this was edited.

I don’t get the change from special feature of obstetrics to job motivation in obstetrics. It does not seem to be the topic the participants are referring to.

Authors:

We deleted the results referring to job motivation from the table and the further descriptions/ discussion.

Summaries of the tables 3.1 and 3.2 are mainly about the differences in perspectives, rather than a summary of the content of these perspectives. What is the point in telling everything is different? We can read that ourselves.

Authors:

We deleted both summaries.

Note on p. 11 line 285 unnecessary

Authors:

Thank you, we deleted this note.